# Modeling of Particulate Pollutants Using a Memory-Based Recurrent Neural Network Implemented on an FPGA

**DOI:** 10.3390/mi14091804

**Published:** 2023-09-21

**Authors:** Julio Alberto Ramírez-Montañez, Jose de Jesús Rangel-Magdaleno, Marco Antonio Aceves-Fernández, Juan Manuel Ramos-Arreguín

**Affiliations:** 1Facultad de Ingeniería, Universidad Autónoma de Querétaro, Querétaro 76010, Mexico; 2Digital Systems Group, Electronics Department, National Institute for Astrophysics, Optics and Electronics, Puebla 72840, Mexico

**Keywords:** air pollution, FPGA, recurrent neural networks, modeling, criteria pollutants

## Abstract

The present work describes the training and subsequent implementation on an FPGA board of an LSTM neural network for the modeling and prediction of the exceedances of criteria pollutants such as nitrogen dioxide (NO_2_), carbon monoxide (CO), and particulate matter (PM_10_ and PM_2.5_). Understanding the behavior of pollutants and assessing air quality in specific geographical regions is crucial. Overexposure to these pollutants can cause harm to both natural ecosystems and living organisms, including humans. Therefore, it is essential to develop a solution that can accurately evaluate pollution levels. One potential approach is to implement a modified LSTM neural network on an FPGA board. This implementation obtained an 11% improvement compared to the original LSTM network, demonstrating that the proposed architecture is able to maintain its functionality despite reducing the number of neurons in its initial layers. It shows the feasibility of integrating a prediction network into a limited system such as an FPGA board, but easily coupled to a different system. Importantly, this implementation does not compromise the prediction accuracy for both 24 h and 72 h time frames, highlighting an opportunity for further enhancement and refinement.

## 1. Introduction

Environmental pollution is a mixture of naturally and anthropogenically produced pollutants [1]. Reducing pollution or maintaining its level is one of the main objectives worldwide to ensure the health of the population and ecosystems [2]. There are different types of classification of environmental pollution, such as water pollution, land pollution, and air pollution [1,2]. The latter has subclassifications such as greenhouse gases, short-lived pollutants, ozone-depleting substances, and finally, criteria pollutants [1,2,3,4].

Monitoring networks were developed to record the behavior of the different criteria pollutants and various environmental conditions [1,5]. The stations are strategically placed across a specific geographic area and equipped with suitable sensors. This arrangement enables the generation of a comprehensive database that contains records of past pollutant behavior. By analyzing these data, we can assess the region’s progress and determine whether certain stations need to be maintained, replaced, or new ones added to the network [5,6].

Developing algorithms based on artificial intelligence has allowed us to interpret and classify large datasets, generating mathematical models of its behavior [7,8,9,10,11] and, at the same time, generating predictive models [12,13,14,15]. Artificial intelligence has two essential areas: machine learning and deep learning [16]. The latter classifies algorithms such as deep neural networks: convolutional neural networks and recurrent neural networks, among others.

Recurrent neural networks are one of the main algorithms implemented to detect a series of behaviors in continuous records [7,10,11,12,13,14,15], maintaining consistency between the previous and new data that this network proposes, emulating the behavior of the analyzed records.

In the present work, an LSTM architecture was implemented, which will be detailed in a later section. The selection of this architecture is based on the wide implementation in registers whose behavior is highly nonlinear [11,15], that is to say that they do not have a periodic behavior, and this behavior is the result of the intervention of different variables. However, this behavior presents certain behaviors that an RNN can detect and model, allowing the generation of a predictive model.

### 1.1. Criteria Pollutants

Criteria pollutants are so named because they are the primary focus of public assessments in documents pertaining to air quality [3,17]. These pollutants are subject to specific concentration limits in the environment to safeguard and uphold the population’s well-being [17,18].

They pertain to a distinct category of atmospheric pollution primarily attributed to human activities, particularly industrial processes [3]. The specific pollutants under consideration include sulfur dioxide, nitrogen dioxide, carbon monoxide, ozone, and particulate matter with diameters less than 10 and 2.5 μm [3,18]. The characteristics of these contaminants are outlined in Table 1 [4,17].

Criteria pollutants have different behaviors; each one is affected by different variables, both climatic and the result of industrial, commercial, or population processes [3,4]. This generates a highly nonlinear behavior, as shown in Figure 1. The acronyms CUT, FAC, HGM, BJU, ATI, CAM, and CCA represent the monitoring stations from which the behavioral records were retrieved [5].

In the present work, we will focus on three criteria pollutants for the modeling stage and a fourth for the prediction stage. In the modeling stage, it will be carbon monoxide (CO) and particulate matter (PM_10_ and PM_2.5_), whose behaviors present a greater randomness in the records and mainly because they have the largest number of continuous records. In the prediction stage, we added ozone (O_3_) as a control criteria pollutant, since it is one of the most-widely studied [3,4].

### 1.2. Monitoring Networks

Atmospheric monitoring is a set of actions that allow measuring the values of meteorological and air quality parameters in a given region. According to the National Institute of Ecology and Climate Change (INECC by its Spanish acronym), atmospheric monitoring in Mexico is used as an instrument for the establishment of environmental policies to protect the health of the population and ecosystems [2,19].

Atmospheric monitoring stations play a crucial role in generating accurate data and aiding in developing air quality standards while ensuring their adherence [20]. These stations have various sensors that record important climatic variables and pollutant concentrations. The primary sensors include temperature, relative humidity, wind direction, and wind intensity, which are essential for monitoring climatic conditions [2,6]. Additionally, pollutant sensors are utilized to measure concentrations of ozone, nitrogen dioxide, nitrogen monoxide, nitrogen oxides, carbon monoxide, sulfur dioxide, PM_10_, and PM_2.5_ [5].

### 1.3. Recurrent Neural Networks

Recurrent neural networks are a deep learning tools that recursively compute new states by applying transfer functions to previous states and their inputs [21,22]. Transfer functions usually comprise an affine transformation followed by a nonlinear function determined by the nature of the problem at hand [10,21].

In 2007, Maass showed that RNNs possess the so-called universal approximation property, which establishes the ability to approximate arbitrary nonlinear dynamical systems with some arbitrary accuracy by performing complex mappings from input sequences to output sequences [12,23].

RNNs do not have a defined layer structure, which allows arbitrary connections between neurons, allowing the creation of a certain temporality, generating a network with memory. There are several types of recurrent networks depending on the number of layers and the way in which backpropagation is performed [12].

Some of the main applications of RNNs are natural language recognition, such as for chatbots or translators, pattern recognition, and prediction in continuous records [10,11,12].

#### LSTM

The long short-term memory (LSTM) recurrent neural network, proposed by Hochreiter and Schmidhunder in 1997, is part of deep learning [14,22,24], widely used in natural language processing and time series [7,11,15].

It is unlike a simple recurrent neural network, which forms a long-term memory in the form of weights between neurons, which are modified during the training of the network, and a short-term memory, defined in the activation functions between the communication of the neuron nodes [12,15,22].

The LSTM model introduces an internal memory block, composed of simple blocks connected in a specific way (see Figure 2a), each of which is described in Equations (Equation 1)–(Equation 6) [22], while in Figure 2b, the neuron to neuron connection and data flow are observed.
Figure 2LSTM network structures (adapted from [15]).
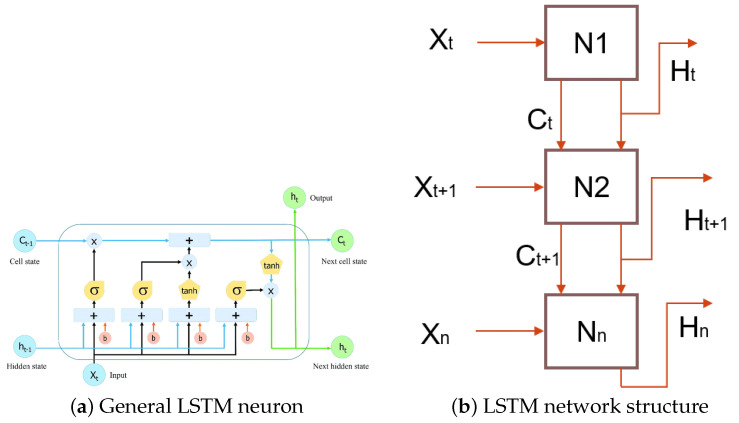

(1)$$f_{t}=\sigma \left(U_{f}*h_{t-1}+W_{f}*x_{t}+b_{f}\right)$$
(2)$$i_{t}=\sigma \left(U_{i}*h_{t-1}+W_{i}*x_{t}+b_{i}\right)$$
(3)$$nc_{t}=tant\left(U_{nc}*h_{t-1}+W_{nc}*x_{t}+b_{nc}\right)$$
(4)$$c_{t}=f_{t}*c_{t-1}+i_{t}*nc_{t}$$
(5)$$h_{t}=o_{t}*tant\left(c_{t}\right)$$
(6)$$o_{t}=\sigma \left(U_{o}*h_{t-1}+W_{o}*x_{t}+b_{o}\right)$$

The variables are described below:

$x_{t}$: initial data (input).

$C_{t-1},C_{t}$: cell state, next cell state.

$h_{t-1},h_{t}$: hidden state, next hidden state/output.

$U_{f},U_{i},U_{nc},U_{o}$: feedback weights.

$W_{f},W_{i},W_{nc},W_{o}$: internal weights.

$b_{f},b_{i},b_{nc},b_{o}$: bias.

### 1.4. FPGA Board

Field programmable gate arrays (FPGAs) are programmable electronic devices based on a matrix of logic blocks whose interconnection and functionality can be configured by the user using a specialized description language (such as VHDL and Verilog) [25,26], belonging to a classification of an embedded system [27]. The term embedded system refers to small systems with limited resources, referencing the operability of a personal computer [25,26,27].

The main advantage of using FPGA cards is that they can be used to implement massively parallel data-processing algorithms, through a programmable structure based on blocks that operate together, allowing a reconfiguration if necessary to improve connections from stage to stage, necessary for a specific algorithm [27,28]. Microcontrollers are another tool used, but they work sequentially, limiting their development process and application to already-defined tasks [27].

FPGA boards contain logic gates, clock controllers, and RAM and ROM memories, among other elements. In the development of this work, we used the DE10-Standard board, which included a Cyclone V SX SoC-5CSXFC6D6F31C6NN, a 925 MHz Dual-Core ARM processor, and a 64 MB SDRAM memory, including GPIO connection ports.

## 2. Methodology

Figure 3 shows a diagram of the implemented methodology, distinguishing the two main stages: the implementation stage at the software and hardware level. These have internal sections, and each one will be explained in more detail in the following sections.

## 3. Data Preparation

The selected database corresponds to the records of the Automatic Atmospheric Monitoring Network (RAMA) of Mexico City, corresponding to the period from 2000 to 2019. The amount of atypical data corresponding to an erroneous measurement, represented in this database by −99, was counted to maintain continuity in the records. Table 2 shows the percentages of invalid data for different monitoring stations whose records correspond to the period mentioned above. The MER station was selected for the development of the present work.

Once the stations with the most valid records have been identified, the next step was to propose the behavior of the missing data, for which the multiple imputation by chained equations (MICE) algorithm was implemented. This algorithm has demonstrated its ability to emulate the behavior of missing data by comparing the valid records [29].

The MICE algorithm operates iteratively, improving the missing data estimates. The base algorithm is described in Algorithm 1. In the first iteration, a simple imputation (mean), then a regression are applied between consecutive pairs until the defined number of iterations is completed [14,15].
**Algorithm 1** MICE algorithm.Fill in missing values from random draws of non-missing data**for** each iteration **do**    **for** each variable v with missing values **do**        Subset data, where v was originally nonmissing        Train model v∼X, where **X** are the other variables in the dataset        Do either:        (1) Replace missing values with predictions from model        (2) Replace missing values using mean matching    **end for****end for**

It is still necessary to consider that, when proposing the behavior of the data in the periods where invalid data were recorded, these only represent an approximate behavior [14]. We cannot ensure that the behavior represented is similar to the one existing in those periods of time [15,29].

## 4. Network Training

The LSTM network that was chosen comprises two main stages. The first stage is the behavioral modeling stage, which consists of three layers. The second stage is the exceedance detection stage, which also consists of three layers. Figure 4 shows the specific number and type of neurons present in each layer. It is important to note that, in Layer 4, the number of neurons is denoted by “x” because it varies depending on the desired time of anticipation for generating the detection. Specifically, it is equivalent to having one neuron per previous hour. For instance, if a 24 h anticipation is required, then 24 neurons are necessary. On the other hand, if the objective is to predict with a 72 h anticipation, then 72 neurons are needed [14].

In the modeling phase of the applied network architecture, the entire dataset was utilized, with 80% of the data designated for training purposes and the remaining 20% reserved for validation. To ascertain the network’s ability to identify behavioral patterns, the validation was conducted using the correlation coefficient (CC) [30] and the root-mean-squared error (RMSE) [30,31]. The mathematical expressions for these metrics are provided below:(7)$$CC=\frac{\frac{\sum_{i=1}^{n}\left(M_{i}-\bar{M}\right)\left(R_{i}-\bar{R}\right)}{n-1}}{\sqrt{\frac{\sum_{i=1}^{n}\left(M_{i}-\bar{M}\right){}^{2}}{n-1}*\frac{\sum_{i=1}^{n}\left(R_{i}-\bar{R}\right){}^{2}}{n-1}}}$$
(8)$$RMSE=\sqrt{\frac{1}{n}\sum_{i=1}^{n}\left(M_{i}-R_{i}\right){}^{2}}$$

The variables are described below:

$R_{i}$: real data.

$\bar{R}$: average of real data.

$M_{i}$: data modeled by the LSTM network.

$\bar{M}$: average of data modeled by the LSTM network.

*n*: total data.

In the section of the detection of exceedances, the first step is to identify what is an exceedance, and for this we used the Mexican standards: NOM-025-SSA1-2021, NOM-021-SSA1-2020, and NOM-022-SSA1-2019. These indicate the maximum permitted value and the period in which it is evaluated (see Table 3) [32].

Since the values to be evaluated are an average of the 8 h and 24 h records, we have a daily value, with which we will be able to identify the previous behavior so that we can label and save the days with exceedances and those that do not with the previous behavior, either 24 or 72 h in advance.

Once the days with and without exceedances have been sorted, they are placed in chronological order, selecting 80% of the data to train the exceedance detection network, while the remaining 20% will be used to evaluate the network.

## 5. Reduction of LSTM Network Parameters

The utilization of neural networks in FPGAs involves employing smaller networks with a complete connection configuration [27,28]. When implementing LSTM networks, individual neurons are constructed as separate blocks [33]. However, creating a complete network within an FPGA becomes challenging due to the constrained resources available on FPGA boards.

Considering the specific architecture of the implemented network, which encompasses 14,578,523 internal parameters, significant memory capacity is necessary solely for storing the weights and biases of each neuron. This calculation does not even take into account the memory space required for storing the intermediate results generated at each stage until the final output is obtained.

Table 4 shows the initial architecture of the implemented LSTM recurrent network and the characteristics of the final architecture. This reduction was obtained by taking as a reference the work of Kaushik Bhattacharya [34], which consists of defining the number of neurons in the initial layer equal to the length of the vector of the variables to be estimated, in this case, 24 times the hours of recording of a day. For the hidden layers, multiplies of the number of neurons in the initial layer were used, using 24 in the same way; if this does not generate similar results to the initial architecture, the number of neurons will be constantly increased or another hidden layer will be added.

In the prediction stage, being a simple architecture, it is not modified, only in the initial layer denoted with an X, since the number of neurons depends on the time with which we want to make the prediction, 24 or 72, referring to the hours of anticipation.

## 6. Implementation of the LSTM Network on an FPGA

As the neural network was previously trained, the weights matrices corresponding to each layer of the network were extracted, to identify the maximum and minimum values and define their conversion to a binary system: implementing two representation options (12 and 16 bit) in a five-point seven and five-point eleven format, respectively; using the most-significant bit as the sign bit, as shown in the Table 5.

The subsequent phase involved the organization of LSTM neuron operations, aiming to exert control over the selection of corresponding weights and the sequential execution of operations while retaining the results for subsequent iterations. Regarding the internal activation functions of the LSTM neuron, an approximation method was applied. Specifically, the sigmoid function was approximated using a polynomial function (Equation (Equation 9)), whereas the hyperbolic tangent function was approximated using a piecewise function (Equation (Equation 10)). Figure 5 shows the representation of the digital design resulting from the development of an LSTM neuron.
(9)$$y=-0.0127x^{3}+0.2419x+0.4999$$
(10)$$y=\left\{\begin{array}{cc} x\geq 1.5 & y=0.97\\ -1.5<x<1.5 & y=-0.1045x^{3}+0.8644x\\ x\leq -1.5 & y=-0.98 \end{array}\right.$$

Figure 6 shows the basic structure of the internal operations of the LSTM neuron, using multiply–accumulate (MAC). This same structure was implemented to represent the dense neurons that symbolize perceptrons. This is shown in Equation (Equation 11).
Figure 6Base operational structure.
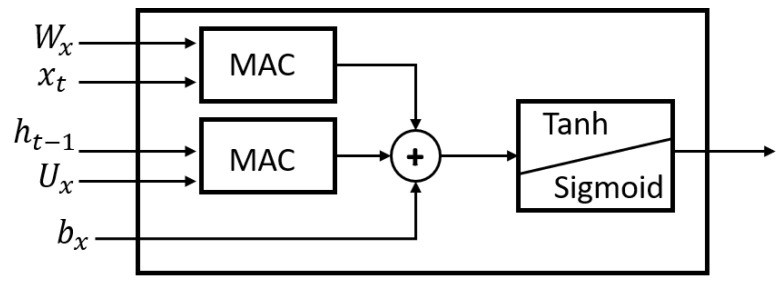

(11)$$y=\sum_{i=1}^{n}\left(x_{i}W_{i}\right)+b$$

The variables are described below:

$x_{i}$: input data.

$W_{i}$: weights.

*b*: bias.

Multiplication operations are characteristic of increasing the number of bits by a factor of two compared to the input operators. This study utilized two fixed-point configurations, one with 12 bit and the other with 16 bit. Consequently, the output of these configurations would be 24 bit and 32 bit, respectively. In order to maintain the same word length, specific datasets were chosen for analysis, covering Bit Positions 7 to 18 and 11 to 26, respectively. The implemented multiplication and addition structure is illustrated in Figure 7. On the other hand, the addition operation preserves the word length both at the output and input, as it is a direct operation.

With the weights defined in the network training, we grouped the corresponding matrices in RAM blocks from a computer to the FPGA through the RS232 communication protocol, synchronizing the memory blocks with the operations to be performed by means of a counter and a second one to define the layer in which it is operating, until finishing the evaluation of the whole network.

The final configuration is depicted in Figure 8, illustrating the RAM memory blocks wherein the weights and bias values are stored. Additionally, it demonstrates the interconnectedness between the blocks, representing each network layer. Internally, these blocks are constituted by components that symbolize the LSTM neuron, as depicted in Figure 5.

## 7. Results and Discussions

To ensure that the reduction of the parameters due to the reduction of LSTM neurons in the second network architecture maintains a similar behavior in the RMSE and CC, we performed evaluations with identical data sections in ten independent tests. We obtained the results shown in Table 6, showing that the modeling proposed by the networks remains stable evaluating neuron by neuron, the initial and final layer, even when evaluated on the computer.

Once the correct operation of the reduced network has been demonstrated, the next step is to compare the percentage error between the initial data and the data to be supplied to the FPGA in 12 bit and 16 bit format and evaluate the accumulated neuron-to-neuron error and the final error.

In Table 7, it is observed that the accumulated error in the minimum final layer belongs to the use of 16 bit; in the same way it, is observed that the error of the initial layer oscillates between 9% and 15%, observing the percentage of error of the first neuron and the twelfth neuron and the accumulated error of the first layer. On the contrary, when observing the behavior of the errors with the 12 bit format, an increase in the accumulated errors is perceived and how it grows up to 25% in the final layer, generating deficient modeling.

Figure 9 shows the level of repeatability of the accumulated errors in the modeling stage in its final layer after 20 tests evaluating different data represented in 12 and 16 bit. It shows that the accumulated error with 16 bit remains constant in a range of 11% to 13% error, while with a 12 bit representation, the error ranges between 25% and 27%.

Figure 10a,b show the behavior between the real data (gray color), the data modeled by the original LSTM network (blue color), and the data modeled by the FPGA architecture (orange color), during the first 72 h (3 days) of January 2022 and 2018, respectively, for PM_10_. It can be seen that the LSTM network architectures (both the original and reduced) have similar behavior, obtaining better results, as shown in Figure 10c for 2018, while in 2022, the networks failed to model the behavior until after the first 56 h. This may be due to the sharp increases recorded and the behavior of the previous year; on the contrary, 2018 presents a better modeling as it has a more-stable behavior.

Figure 10c shows an increase in the first 12 h of January 2018, showing the difference in the modeling between both LSTM neural network architectures.

Figure 11a,b show the behavior between the real data (gray color), the data modeled by the original LSTM network (blue color), and the data modeled by the FPGA architecture (orange color), during the first 72 h (3 days) of January 2022 and 2018, respectively, for PM_2.5_. As in the previous figure, it is observed that the LSTM network architectures (both original and reduced) have a similar behavior. In the context of high peak modeling (in the presence of outliers), a notable issue arises due to the dissimilar behavior exhibited by various architectures. Unlike PM_10_, which display abrupt changes, PM_2.5_ demonstrate a more-stable behavior, thereby setting them apart.

In Figure 11c is an increase in the first 12 h of January 2018, showing the difference in the modeling between both LSTM neural network architectures.

Figure 12a,b show the behavior of carbon dioxide (CO), which presents a more-chaotic behavior than the behavior of PM_10_ and PM_2.5_; shown is the behavior of the real data (gray color), the data modeled by the original LSTM network (blue color), and the data modeled by the FPGA architecture (orange color), during the first 72 h (3 days) of January 2022 and 2018, respectively. Given that this behavior was registered during the training of the different architectures, it is possible to emulate the behavior better; clearly, there are some points where the network registers higher or lower than the real ones.

Figure 12c shows an increase in the first 12 h of January 2018, showing the difference in modeling between both LSTM neural network architectures.

Figure 13a,b show the behavior of ozone dioxide (O_3_), which presents a simpler behavior than the behavior of PM_10_ and PM_2.5_; shown is the behavior of the real data (gray color), the data modeled by the original LSTM network (blue color), and the data modeled by the FPGA architecture (orange color), during the first 72 h (3 days) of January 2022 and 2018, respectively. Given that this behavior was registered during the training of the different architectures, it is possible to emulate the behavior better; clearly, there are some points that the network registers higher or lower than the real ones.

Figure 13c shows the behavior in the first 12 h of January 2018, showing the difference in modeling between both LSTM neural network architectures.

Finally, Figure 14 shows the evaluation of the prediction stage, for which four pollutants (PM_10_ PM_2.5_, O_3_, CO) were used at two time intervals (24 and 72 h in advance). In order to guarantee the robustness of the network architecture, the results at 24 h of prediction were more compact for PM_10_ (85–89%), PM_2.5_ (81–84%), and CO (77–82%); in the case of O_3_, it had the highest amplitude from 81–92%; they had atypical points, but these did not decrease more than 73% in the case of CO.

On the other hand, if an exceedance is predicted 72 h in advance, the valid percentages increase their range, going from 80 to 90% for PM_10_, 75 to 83% for PM_2.5_, for O_3_ from 80 to 90%, and for CO from 65 to 80%, showing an atypical drop of 50%. Of the four criteria pollutants evaluated, CO is the one that presented the greatest problem for the detection of exceedances.

## 8. Conclusions

In the implementation of a neural network, in this case, an LSTM architecture previously trained on a computer, it is necessary to know the amount of internal parameters since the available memory capacity is a factor to be taken into account in an FPGA board. Therefore, the process of reducing the network is a methodological process, where it is necessary to evaluate stage by stage, verifying that the accumulated error does not tend to grow.

The limitation of memory on an FPGA board is a negative factor, which is compensated by the ease of connection of different peripheral devices, in this case, environmental sensors. The adaptability of the FPGA board allows us to determine the necessary logic requirements for the implementation of the binary values of the weight matrices of the implemented network architecture.

When converting decimal values to binary, it is imperative to carefully consider both the desired accuracy (resolution) and the fundamental integer representation. Neglecting these aspects in internal operations can lead to a significant loss of information, not to mention the criticality of preserving a sign bit.

The inherent characteristics of the network enabled the maintenance of a consistent error level at each stage. When comparing this operation with the original architecture of the LSTM neural network, the error remained within a range of about 11% during the modeling stage. Furthermore, in the prediction stage, the accumulated errors did not significantly differ between the two evaluated architectures.

One issue that requires attention in future endeavors is identifying the precise moment when a contaminant surpasses the acceptable limit and enhancing the prediction accuracy for longer time frames.

## Figures and Tables

**Figure 1 micromachines-14-01804-f001:**
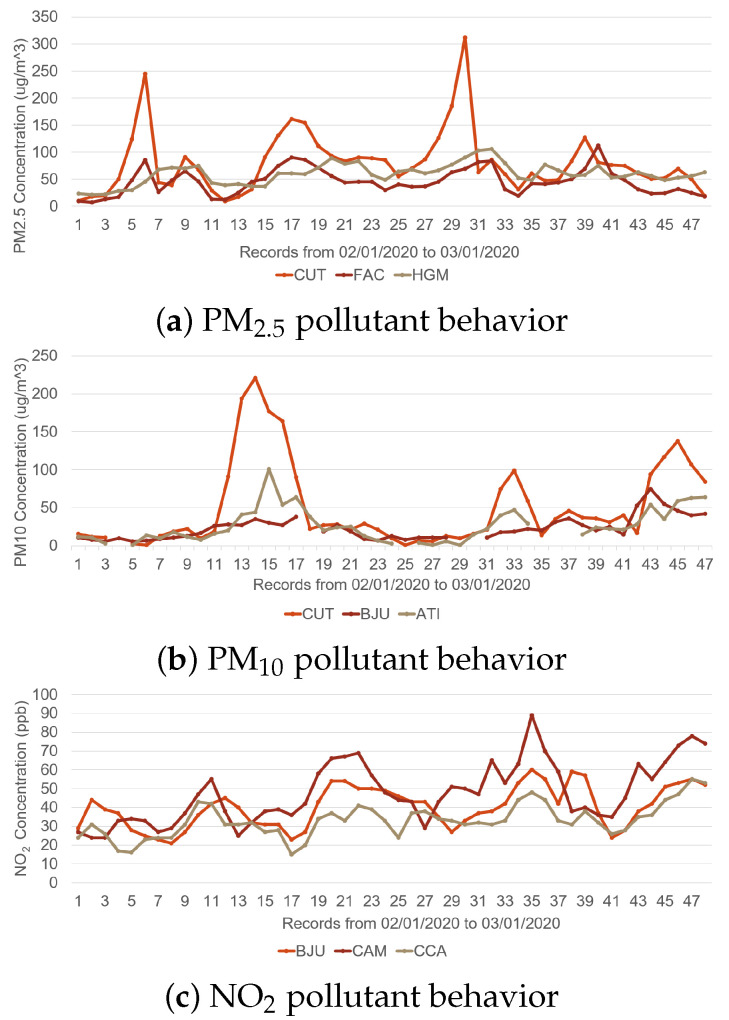
Comparison of recorded behavior for different criteria pollutants. (**a**) PM_2.5_, (**b**) PM_10_ and (**c**) NO_2_.

**Figure 3 micromachines-14-01804-f003:**
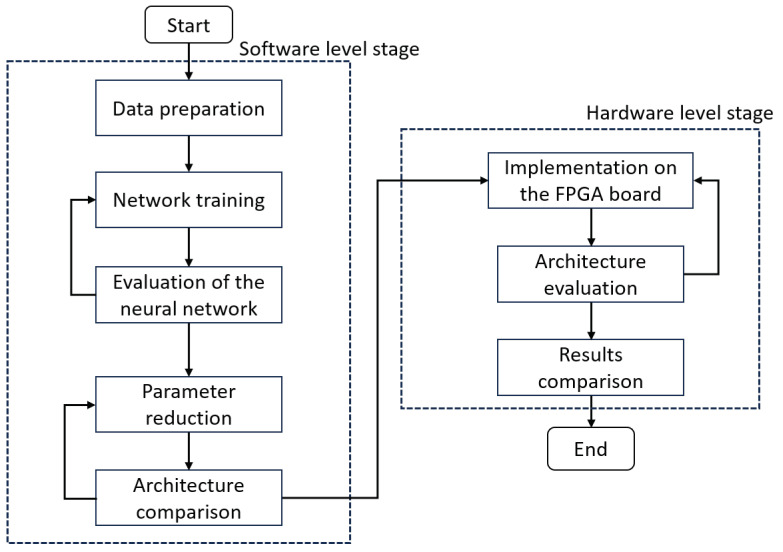
Methodology.

**Figure 4 micromachines-14-01804-f004:**
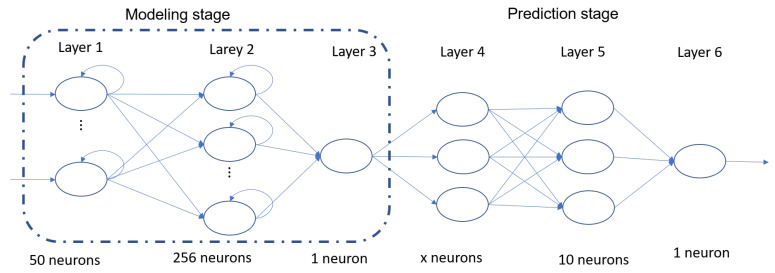
Stages for proposed LSTM structure.

**Figure 5 micromachines-14-01804-f005:**
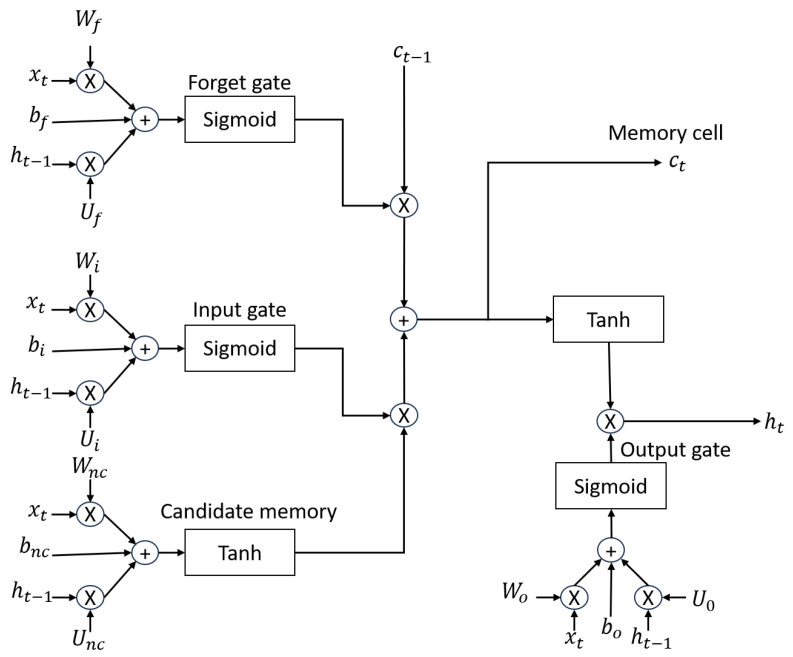
LSTM neuron in VHDL.

**Figure 7 micromachines-14-01804-f007:**
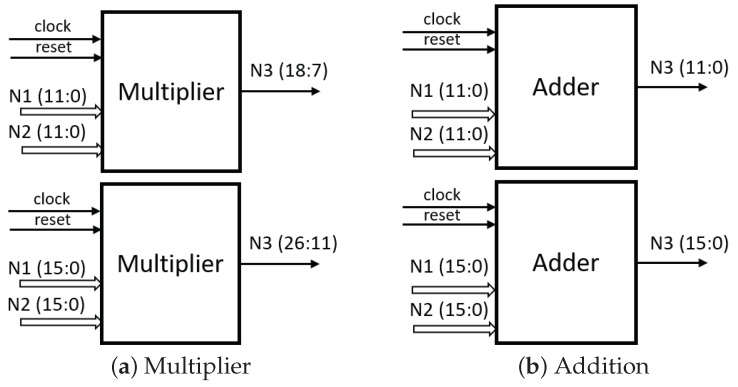
LSTM network structures.

**Figure 8 micromachines-14-01804-f008:**
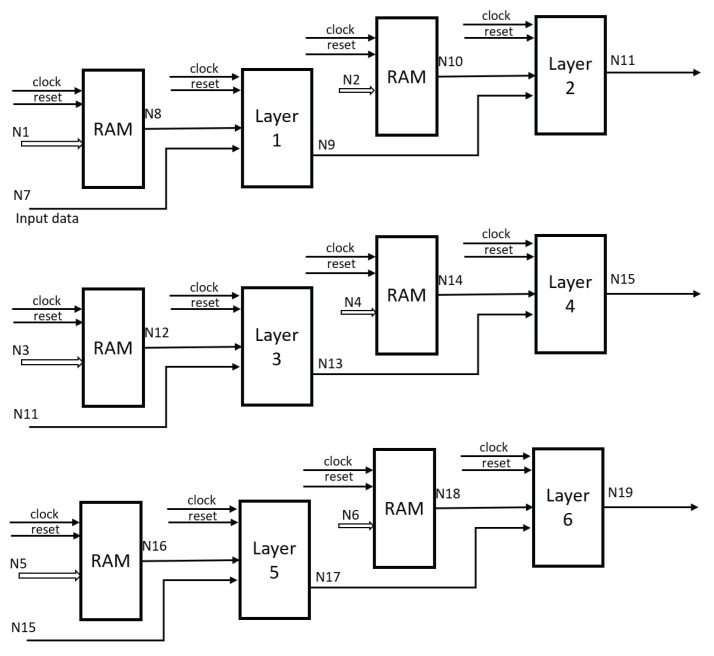
Final structure.

**Figure 9 micromachines-14-01804-f009:**
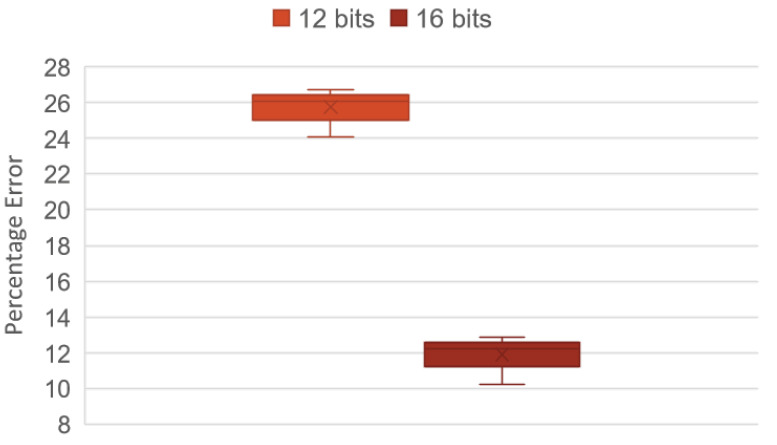
Percentage error.

**Figure 10 micromachines-14-01804-f010:**
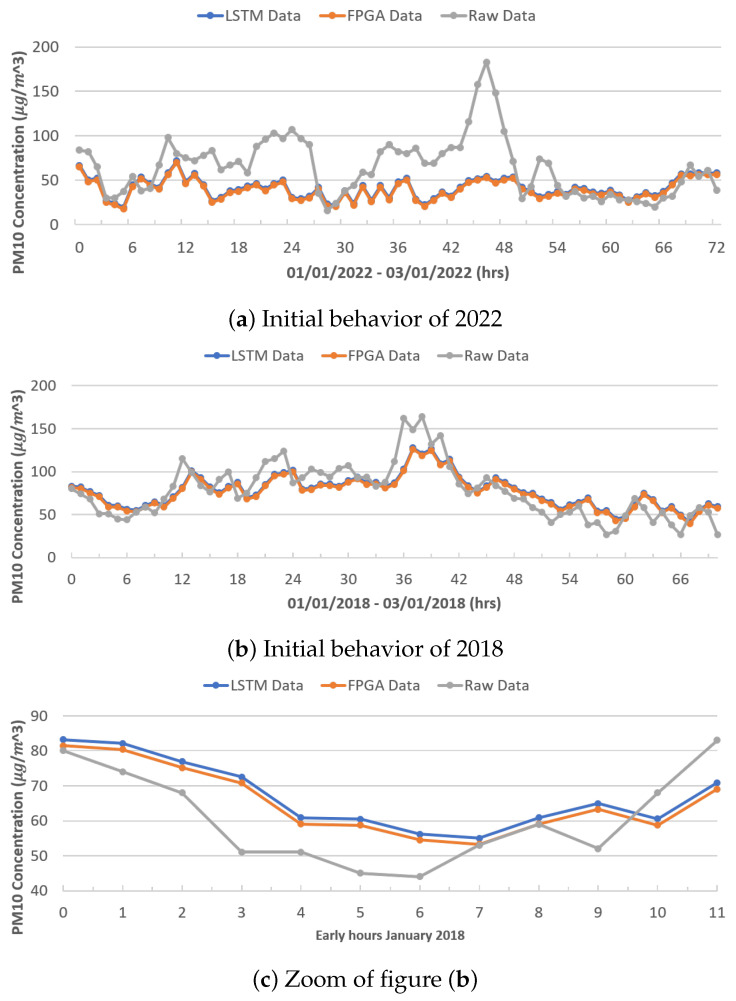
Modeling results.

**Figure 11 micromachines-14-01804-f011:**
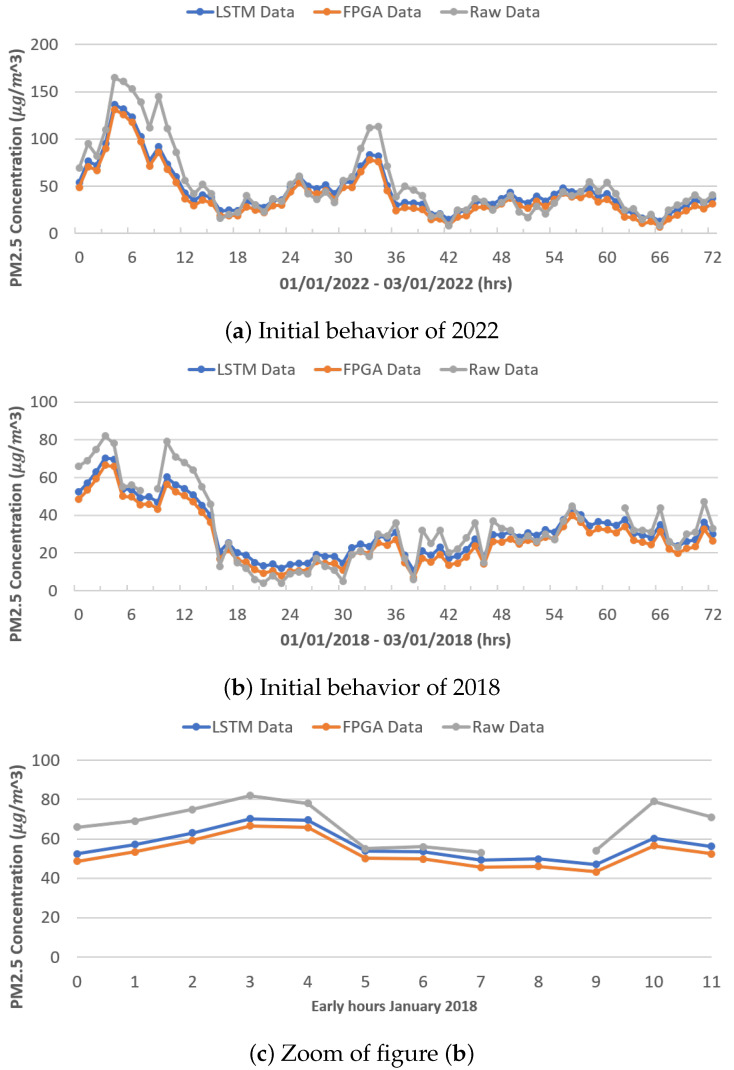
Modeling results.

**Figure 12 micromachines-14-01804-f012:**
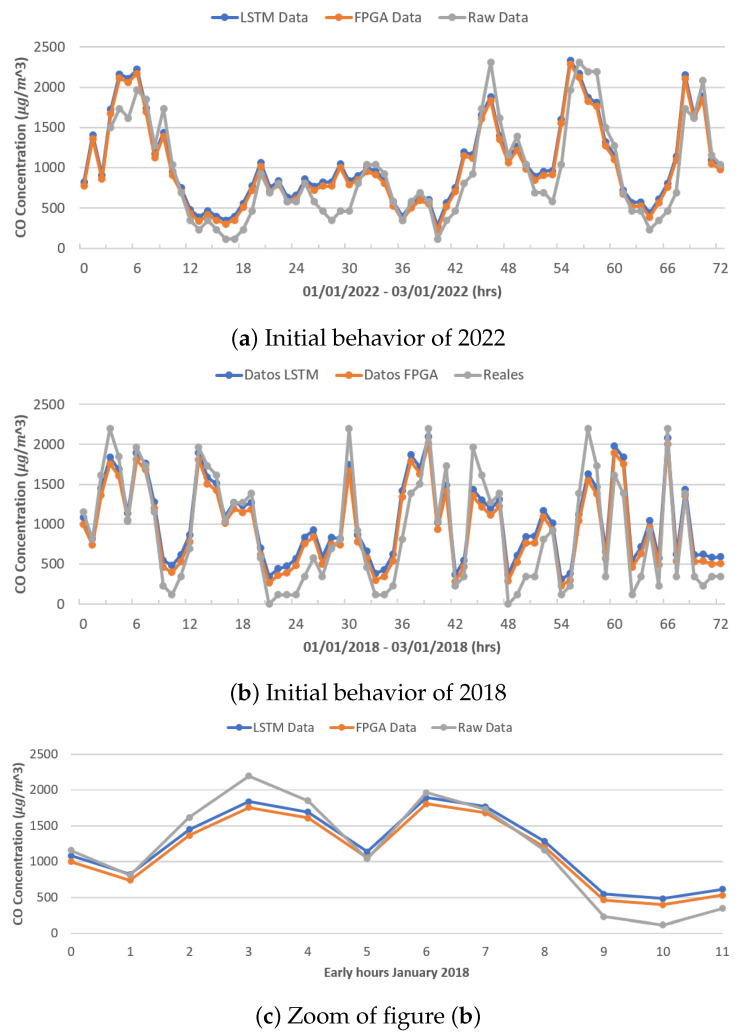
Modeling results.

**Figure 13 micromachines-14-01804-f013:**
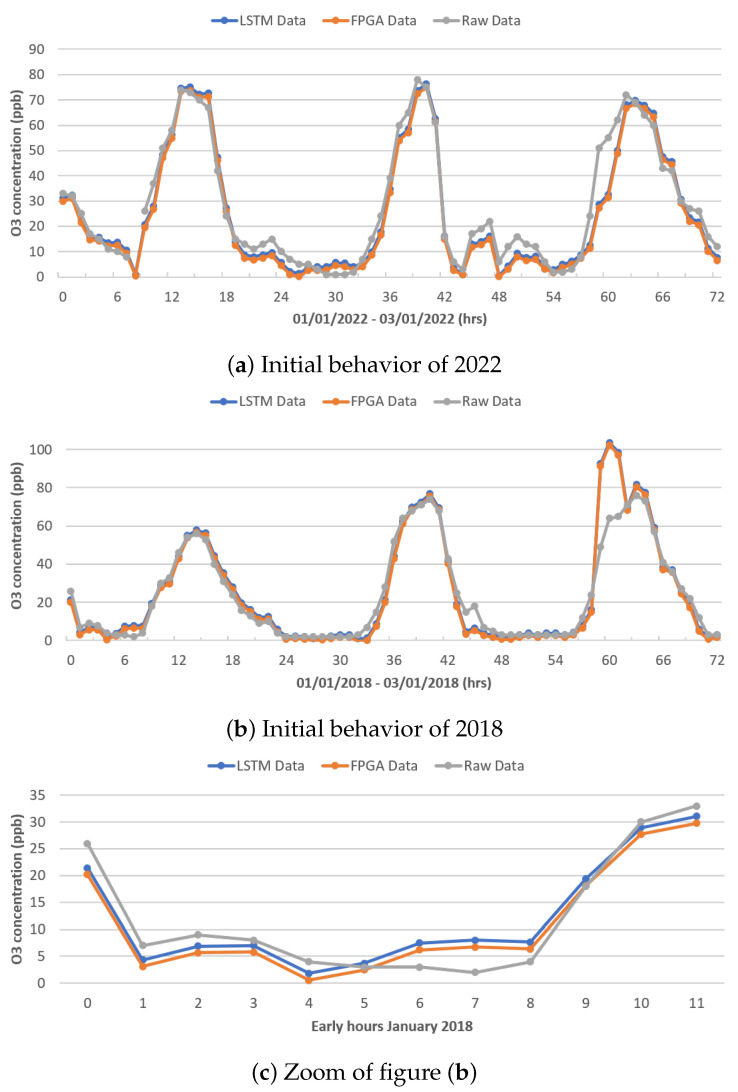
Modeling results.

**Figure 14 micromachines-14-01804-f014:**
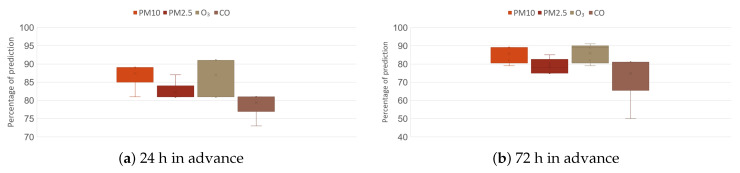
Prediction results.

**Table 1 micromachines-14-01804-t001:** Criteria pollutant characteristics.

Pollutant	Symbol	Characteristics	Damage to Health
Sulfur dioxide	SO_2_	It is a colorless gas with a pungent odor generated by fossil fuel combustion and the smelting of sulfur-containing ores.	Irritating to the respiratory tract. In high concentrations, may cause bronchitis and tracheitis.
Nitrogen dioxide	NO_2_	The main sources of anthropogenic NO_2_ emissions are combustion processes (heating, electricity generation, and vehicle and ship engines).	Irritating to the respiratory tract. In high concentrations, may cause bronchitis and pneumonia.
Carbon monoxide	CO	It is produced from the incomplete combustion of coal. Both human activities and natural sources produce it.	In high concentrations, it disables oxygen transport to cells and cause dizziness, headache, unconsciousness, and even death.
Ozone	O_3_	It is formed by the reaction with sunlight of pollutants such as nitrogen oxides (NOx) from vehicle or industrial emissions and volatile organic compounds emitted by vehicles, solvents, and industry.	Irritating to the respiratory tract. In high concentrations, reduces lung function, worsens asthma, and aggravates chronic lung diseases.
Particulate matter	PM_10_	They are solid or liquid particles of dust, ashes, soot, metallic particles, cement, or pollen, dispersed in the atmosphere and whose diameter is less than 10 μm.	Aggravates asthma and cardiovascular respiratory diseases. Prolonged exposure may increase the risk of mortality.
	PM_2.5_	As it is less than 2.5 μm in size, it remains suspended in the atmosphere for long periods of time, travel long distances, and penetrate the interiors of homes, offices, etc., thus exposing the population to them for longer periods of time.	Aggravates asthma, reduces lung function, and is associated with the development of diabetes.

**Table 2 micromachines-14-01804-t002:** Invalid data percentages.

	MER	CUA	PED	BJU
PM_10_	9%	10%	38%	12%
PM_2.5_	11%	23%	30%	13%
CO	11%	23%	30%	13%
NO_2_	12%	8%	24%	43%

**Table 3 micromachines-14-01804-t003:** Data representative of the Mexican basic standards.

		Maximum	Average
Norm	Pollutant	Value	Time
	Criterion	(${\mathrm{ug}/m}^{3}$)	(h)
NOM-025-SSA1-2021	PM_10_	70	24
NOM-025-SSA1-2021	PM_2.5_	41	24
NOM-022-SSA1-2019	$SO_{2}$	104.8	24
NOM-021-SSA1-2020	CO	10,000	8

**Table 4 micromachines-14-01804-t004:** LSTM neural network architecture.

	Layer 1	Layer 2	Layer 3	Layer 4	Layer 5	Layer 6
	LSTM	LSTM	Dense	Dense	Dense	Dense
Initial	50	256	1	x	10	1
Final	24	24	1	x	10	1

**Table 5 micromachines-14-01804-t005:** Distribution of the fixed-point 12 and 16 bit word.

Sing	Integer Part	Decimal Part
1 bit	4 bit	7 bit
0	0000	0.0000000
1 bit	4 bit	11 bit
0	0000	0.00000000000

**Table 6 micromachines-14-01804-t006:** Comparison of the results obtained.

Neural	Internal	RMSE	CC	Prediction	
	Parameters			24 h	72 h
Initial	325,286	16.145	0.97	90%	85%
Final	7486	18.251	0.96	89%	84%

**Table 7 micromachines-14-01804-t007:** Error comparison.

No. of Bits	Initial	Final Errors			
	Values	Neuron 1	Neuron 12	Layer 1	Final Layer
12	0.0073%	23.17%	24.54%	15.69%	25.75%
16	0.0046%	15.45%	9.27%	10.31%	11.91%

## Data Availability

Not applicable.

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
