# Peer review of "Modeling of Particulate Pollutants Using a Memory-Based Recurrent Neural Network Implemented on an FPGA"

_micromachines, 2023, doi:10.3390/mi14091804_

Round 1
Reviewer 1 Report
While the research on implementing a modified LSTM on an FPGA for pollution evaluation is intriguing, the authors' presentation lacks clarity.
- Please clarify the contributions of your research. In the abstract, you mention, “This implementation obtained an approximate error of 11% compared to the original LSTM network.” However, I couldn't grasp the extent of improvement your proposal offers over the original LSTM network. How does your method compare to the original LSTM network? Is there any improvement?
- I could not understand why the implementation does not compromise the prediction accuracy for both 24-hour and 72-hour time frames. It is easy to implement the function.
- Regarding the figures in the manuscript, please ensure that the units on the axes are clearly labeled.
- I strongly suggest breaking up longer sentences into shorter ones, such as the sentences on lines 55-58. This would make the narrative and logic clearer. As it stands, sentences with obscure logic are challenging to follow.
- “Where" should not be used to start a paragraph.
- The structure of the manuscript could benefit from refinement. The 'Introduction' section provides an overview of the research background, highlights unresolved issues, and outlines your research and primary contributions. I recommend reorganizing this section for better clarity.
Actually, writing is hard work. It's not easy to describe one's research with clear logic and precise expression, but it's a basic requirement. It is recommend to thoroughly review and revise the content for clarity and logical flow. Consulting a native English writer might enhance the manuscript's coherence and articulation. Looking forward to your revision.
Revision is necessary.
Author Response
eviewer 1:
-
Please clarify the contributions of your research. In the abstract, you mention, “This implementation obtained an approximate error of 11% compared to the original LSTM network.” However, I couldn't grasp the extent of improvement your proposal offers over the original LSTM network. How does your method compare to the original LSTM network? Is there any improvement?
-
I could not understand why the implementation does not compromise the prediction accuracy for both 24-hour and 72-hour time frames. It is easy to implement the function.
-
Regarding the figures in the manuscript, please ensure that the units on the axes are clearly labeled.
-
I strongly suggest breaking up longer sentences into shorter ones, such as the sentences on lines 55-58. This would make the narrative and logic clearer. As it stands, sentences with obscure logic are challenging to follow.
-
“Where" should not be used to start a paragraph.
-
The structure of the manuscript could benefit from refinement. The 'Introduction' section provides an overview of the research background, highlights unresolved issues, and outlines your research and primary contributions. I recommend reorganizing this section for better clarity.
Response:
I appreciate your feedback. I will now present the revisions that have been applied. In the first instance, adjustments were enacted to underscore that the intention is not solely focused on minimizing the acquired errors. Rather, the primary aim is to uphold the integrity of the behavioral model across various pollutants. This involves the utilization of an LSTM network architecture, which has been previously established, for comparative evaluation against a reengineered iteration deployed on an FPGA board.
In the second point, the selection of time intervals was made with the specific aim of assessing predictive capabilities within actionable time frames. Intervals of 24 and 72 hours were chosen for this purpose. Given the nature of forecasting, the decision was made to exclude longer time intervals.
In regards to points 3 and 5, a comprehensive review was conducted. This encompassed an examination of the presented images and an analysis of the correct usage of the term "where" as an introductory element for paragraphs.
Concerning point 4, modifications were applied to the wording in order to enhance comprehensibility while retaining the essence of the explanatory content.
Lastly, within point 5, the initial structure of the introduction underwent a meticulous revision. Your valuable insights were greatly appreciated, and subsequent efforts were directed towards a comprehensive rewrite of this section, aimed at achieving heightened clarity.
Reviewer 2 Report
After reading the article "Modeling of particulate pollutants using a memory-based recurrent neural network implemented on an FPGA" I give the following conclusion.
Major remarks:
1. It is not completely clear from the paper why the authors chose LSTM, because it is quite an old technology. Such a choice should be justified and compared with new approaches.
2. The size of the dataset and the form in which the data are presented in it are not completely clear. There is no reference to the dataset.
3. It is not clear why the authors realized ANN on FPGA. Is it supposed to be an embedded system? How is it better than a microcontroller board, such as here [https://doi.org/10.1177/0142331221100337, https://doi.org/10.1109/SED51197.2021.9444517, https://doi.org/10.1109/ICICoS51170.2020.9299007]? Few examples of using FPGA for neural network inference are given (additional examples: [https://doi.org/10.1109/ACCESS.2019.2917312, http://doi.org/10.1109/ElConRus51938.2021.9396702, https://doi.org/10.3390/electronics10060681].
The authors have implemented their versions of neurons on neural network resources. How is this better than using off-the-shelf libraries that allow you to generate FPGA firmware based on a high-level implementation of a neural network? (Look for example, OpenVINO [https://doi.org/10.1016/j.sysarc.2021.102062, https://doi.org/10.1109/RusAutoCon52004.2021.9537452, https://doi.org/10.1016/j.jisa.2021.102933] or MATLAB [https://doi.org/10.1093/comjnl/bxac167, https://doi.org/10.1109/TVLSI.2019.2947639, https://doi.org/10.1142/S0129626423500019]).
4. In my opinion, the obtained prediction accuracy results are rather low for this kind of tasks. To convince me in the opposite can only compare with the works of other authors who used similar approaches for a similar problem.
5. The conclusions contain too much superfluous general reasoning not directly related to the results of the paper.
Minor remarks:
1. Line 102 and expressions 4 and 5: in the expressions the letter "c", but in the description "C". Similarly for "x" in formulas 1-3, 6 and line 101. There is no description of what "n" is.
2. Figure 3 should be made more "horizontal" and placed immediately after section 2 (after line 121).
3. Lines 123-124 mention the selected dataset, but no reference to it is given.
4. Line 135 mentions Algorithm 1, which is across the page. It should be moved closer to the first mention. The algorithm itself is rather simple and its description in this form is, in my opinion, not very informative.
5. Line 145 mentions Table 4, which is one page later. It should be moved closer to the first mention or the text should be changed.
6. After line 211. Formulas 9 and 10. It is worth describing in more detail where the values come from.
7. The results (lines 242-245) describe the evaluation of the data on the computer, but there is no data about the network itself on the computer. It is worth describing this a bit.
8. Figure 8 should be replaced.
9. Line 264: instead of "PM10" it should be "PM10".
10. Line 193 and Figure 12. Suddenly another type of O3 pollution is added for which there was no prediction data before.
11. Reference list:
a. Some of the sources have the year in bold and some do not.
b. Not all publications have a doi.
c. Some of the publications have "pp" for page numbers, while the rest do not.
d. Source 5. Line 11. There is an opening bracket that does not close.
e. Source 31. It is worth removing the CAPSLOCK in part of the text.
Author Response
Reviewer 2:
Major remarks:
-
It is not completely clear from the paper why the authors chose LSTM, because it is quite an old technology. Such a choice should be justified and compared with new approaches.
-
The size of the dataset and the form in which the data are presented in it are not completely clear. There is no reference to the dataset.
-
It is not clear why the authors realized ANN on FPGA. Is it supposed to be an embedded system? How is it better than a microcontroller board, such as here [https://doi.org/10.1177/0142331221100337, https://doi.org/10.1109/SED51197.2021.9444517, https://doi.org/10.1109/ICICoS51170.2020.9299007]? Few examples of using FPGA for neural network inference are given (additional examples: [https://doi.org/10.1109/ACCESS.2019.2917312, http://doi.org/10.1109/ElConRus51938.2021.9396702, https://doi.org/10.3390/electronics10060681].
The authors have implemented their versions of neurons on neural network resources. How is this better than using off-the-shelf libraries that allow you to generate FPGA firmware based on a high-level implementation of a neural network? (Look for example, OpenVINO [https://doi.org/10.1016/j.sysarc.2021.102062, https://doi.org/10.1109/RusAutoCon52004.2021.9537452, https://doi.org/10.1016/j.jisa.2021.102933] or MATLAB [https://doi.org/10.1093/comjnl/bxac167, https://doi.org/10.1109/TVLSI.2019.2947639, https://doi.org/10.1142/S0129626423500019]).
-
In my opinion, the obtained prediction accuracy results are rather low for this kind of tasks. To convince me in the opposite can only compare with the works of other authors who used similar approaches for a similar problem.
-
The conclusions contain too much superfluous general reasoning not directly related to the results of the paper.
Response:
Thank you for your feedback. I will now present the revisions that have been undertaken.
Regarding point 1, a revision has been executed to elucidate the rationale behind the selection of an LSTM RNN. This has been done to substantiate its advantages within the context of this particular application. The authors would like to express my gratitude for your valuable input.
With reference to points 2 to 4, the recommended literature has been meticulously reviewed. Consequently, the justification for this manuscript has been refined to provide a more robust rationale for both the methodology employed and the resultant findings.
Lastly, in point 5, the conclusion section has been refined to accentuate the appropriateness of deploying an RRN on an FPGA card for the purpose of environmental monitoring. I extend my appreciation for your insightful observations.
Comments on the Quality of English Language
Minor remarks:
-
Line 102 and expressions 4 and 5: in the expressions the letter "c", but in the description "C". Similarly for "x" in formulas 1-3, 6 and line 101. There is no description of what "n" is.
-
Figure 3 should be made more "horizontal" and placed immediately after section 2 (after line 121).
-
Lines 123-124 mention the selected dataset, but no reference to it is given.
-
Line 135 mentions Algorithm 1, which is across the page. It should be moved closer to the first mention. The algorithm itself is rather simple and its description in this form is, in my opinion, not very informative.
-
Line 145 mentions Table 4, which is one page later. It should be moved closer to the first mention or the text should be changed.
-
After line 211. Formulas 9 and 10. It is worth describing in more detail where the values come from.
-
The results (lines 242-245) describe the evaluation of the data on the computer, but there is no data about the network itself on the computer. It is worth describing this a bit.
-
Figure 8 should be replaced.
-
Line 264: instead of "PM10" it should be "PM10".
-
Line 193 and Figure 12. Suddenly another type of O3 pollution is added for which there was no prediction data before.
-
Reference list:
-
Some of the sources have the year in bold and some do not.
-
Not all publications have a doi.
-
Some of the publications have "pp" for page numbers, while the rest do not.
Round 2
Reviewer 1 Report
The authors has revised the manuscript carefully according to the previous comments. The article could be considered to be accepted after correct the following minor issues:
1. For the sentence "This implementation obtained an error of approximately 11% 9 compared to the original LSTM network", it should be corrected to "This implementation obtained improvement by ... compared to the original LSTM network"
2. "where" should not be used to start a paragraph.
The authors need to check the writings of the manuscript carefully. For example, "atypical" should be "a typical"
Author Response
Response to Reviews
Manuscript ID: micromachines-2577320
Title: Modeling of particulate pollutants using a memory-based recurrent neural network implemented on an FPGA.
Dear Assistant Editor and reviewers, on behalf of the members of the team, I thank you for your comments on the manuscript entitled Modeling of particulate pollutants using a memory-based recurrent neural network implemented on an FPGA. Modeling of particulate pollutants using a memory-based recurrent neural network implemented on an FPGA
Reviewer 1:
The authors has revised the manuscript carefully according to the previous comments. The article could be considered to be accepted after correct the following minor issues:
1. For the sentence "This implementation obtained an error of approximately 11% 9 compared to the original LSTM network", it should be corrected to "This implementation obtained improvement by ... compared to the original LSTM network"
I appreciate your suggestion in advance, which was made to improve the understanding of the objective of the work presented. The correction is located in the abstract section between lines 8 and 10.
2. "where" should not be used to start a paragraph.
I appreciate the comment on the change of the word "where" at the beginning of a paragraph, which was replaced by indicating the characteristics of the variables being described. The correction is located in the “LSTM” section on line 107, in the “Network training” section on line 171, and finally in the “Implementation of the LSTM network on FPGA” section on line 228.
Reviewer 2 Report
First of all, I would like to note a very inconvenient form of the authors' response to the reviewer's comments: a text form instead of a file, which still has to be found. Everything is mixed up.
As a rule, authors should give a paragraph of the reviewers comment, then a response to it and separately - what corrections they made. All this is highlighted in color or in a different font.
Minor remarks were not answered at all.
Also, the method of highlighting chosen by the authors was very unsuccessful. It was very difficult in the new version to understand where the changes were made.
Even though the authors claim to have addressed all 5 points of the remarks, I didn't see it:
- I didn't find where "the revision has been executed to elucidate the rationale behind the selection of an LSTM RNN";
- "the justification for this manuscript has been refined to provide a more robust rationale for both the methodology employed and the resultant findings" didn't happen either;
- the conclusions did not change at all, although the authors claim to have done so.
I did not check whether all minor remarks were addressed, but, for example, the remark about doi and bold year in references remains unsatisfied.
+ Figure 14. has broken away from the rest of the text and is "flying" in the center of page 17.
Author Response
Manuscript ID: micromachines-2577320
Title: Modeling of particulate pollutants using a memory-based recurrent neural network implemented on an FPGA.
Dear Assistant Editor and reviewers, on behalf of the members of the team, I thank you for your comments on the manuscript entitled Modeling of particulate pollutants using a memory-based recurrent neural network implemented on an FPGA. Modeling of particulate pollutants using a memory-based recurrent neural network implemented on an FPGA
Reviewer 2.
First of all, I would like to note a very inconvenient form of the authors' response to the reviewer's comments: a text form instead of a file, which still has to be found. Everything is mixed up.
As a rule, authors should give a paragraph of the reviewers comment, then a response to it and separately - what corrections they made. All this is highlighted in color or in a different font.
Minor remarks were not answered at all.
Also, the method of highlighting chosen by the authors was very unsuccessful. It was very difficult in the new version to understand where the changes were made.
Before starting by indicating the corrections, we are grateful to the reviewer for his comments, and we present a new format for the response.
Even though the authors claim to have addressed all 5 points of the remarks, I didn't see it:
- I didn't find where "the revision has been executed to elucidate the rationale behind the selection of an LSTM RNN";
We appreciate the observations made, the reason for the selection of an LSTM RNN architecture. Briefly describing the qualities of this architecture and why it is useful in the application made.
- "the justification for this manuscript has been refined to provide a more robust rationale for both the methodology employed and the resultant findings" didn't happen either;
We appreciate the comment, since it is an observation in which it seeks to improve the quality of the work presented. In which I would like to confirm that it was modified seeking to enhance the justification of the implementation carried out. The corrected justification is described in the introductory section on lines 40 to 45, conditioning the text so as not to lose clarity.
- the conclusions did not change at all, although the authors claim to have done so.
We appreciate your comments, possibly in seeking to maintain the idea in the conclusion presented were made minimal changes, so it seems to be the same. However, on this occasion a general change was made, we hope this action is distinguished, we appreciate your demand to achieve the desired quality. The specific correction in the conclusion is located in the paragraphs located from lines 331 to 339.
I did not check whether all minor remarks were addressed, but, for example, the remark about doi and bold year in references remains unsatisfied.
We appreciate your observation on the format of the references used, unfortunately we do not know why the format is not homogeneous, since the .bib file was reviewed rigorously but failed to detect any problem, it is a task that we are still pending in the programming of latex templates.
+ Figure 14. has broken away from the rest of the text and is "flying" in the center of page 17.
We appreciate the comment, which was a formatting error in the body of the manuscript, modifying it so that figure 14 is in its ideal position. The correction is located on page 16, locating appropriately the mentioned figure.
Round 3
Reviewer 2 Report
I believe that although the authors responded to the comments, they did so over formally. The substance of the paper has not improved enough to satisfy me (see comments on previous rounds of the review). I see that we have reached an impasse.
If the other reviewers and the editor are content with the article, I will not block it or object to its publication.
I see no point in continuing to review it, since the authors cannot improve the article, and no matter how hard I try to tell the authors what is wrong with it, it is useless. In this case I am powerless, let it be published as it is.
Described in previous rounds of the review.